A hybrid deep learning approach with progressive cyclical CNN and firebug swarm optimization for breast cancer detection

Jakkaladiki Sudha Prathyusha 1 sudha.jakkaladiki@uhk.cz
Malý Filip 2
1 Department of Informatics and Quant. Methods, University of Education Hradec Kralove , Hradec Králové , Czech Republic
2 Department of Informatics and Quant. Methods, University of Hradec Králové , Hradec Králové , Czech Republic
Chicco Davide
Electronic publication date: 2025 Sep 11
Publication date: 2025
Volume: 11
Electronic Location ID: e3119
Received 2025 Jan 10; Accepted 2025 Jul 21
Copyright: © 2025 Jakkaladiki and Malý
Copyright year: 2025
Copyright holder: Jakkaladiki and Malý
License: This is an open access article distributed under the terms of the Creative Commons Attribution License, which permits unrestricted use, distribution, reproduction and adaptation in any medium and for any purpose provided that it is properly attributed. For attribution, the original author(s), title, publication source (PeerJ Computer Science) and either DOI or URL of the article must be cited.
License URL: https://creativecommons.org/licenses/by/4.0/

Keywords: Breast cancer detection, Mammography, Deep learning, Hybrid feature extraction, Firebug swarm optimization, Progressive cyclical CNN

Funding: Excellence project run at the Faculty of Informatics and Management, University of Hradec Kralove, Czech Republic This work was supported by the Excellence project run at the Faculty of Informatics and Management, University of Hradec Kralove, Czech Republic. The funders had no role in study design, data collection and analysis, decision to publish, or preparation of the manuscript.

==============================
The practice of diagnosing breast cancer retains its scope for improvement in medical imaging, where every correct and timely diagnosis enhances the survival rate of patients. This article presents an integrated approach utilizing patch-wise breast image segmentation, hybrid deep feature extraction, followed by progressive cyclical convolutional neural networks (P-CycCNN), and firebug swarm optimization (FSO) to enhance breast cancer detection. This method first incorporates image segmentation by patches to break down the mammography images into smaller patches, which are easier to focus on and allow for the extraction of more features to boost detection rates. Hybrid feature extraction combines convolutional neural network (CNN) features extracted from pre-trained models with handcrafted features that describe texture and shape, thereby enabling the model to grasp the nuances of both coarse and fine images comprehensively. The progressive cyclical CNN strategy incorporates cyclical, re-adjusted learning rates and a progressive training schedule to accelerate and enhance the model’s convergence. FSO is introduced to adjust the hyperparameters of the CNN topology, including the learning rate and regularisation parameters, thereby enhancing training and feature-fusion processes. Evaluated on the Curated Breast Imaging Subset of the Digital Database for Screening Mammography (CBIS-DDSM) dataset, the proposed model achieved 98% test accuracy, 95% precision, 97.2% recall, 96% F1-score, and an AUC of 0.95, outperforming baseline CNN models by 4%–6% across key metrics. This approach holds great potential for enhancing detection systems in clinics, allowing earlier and more accurate detection of malignant lesions.

Introduction

The field of pathology plays a pivotal role in delivering accurate disease diagnoses, which directly impact treatment decisions and patient outcomes. Since the mid-19th century, the microscope has been the primary diagnostic tool (Ackerknecht, 1953). Despite advancements in computer-assisted image analysis, such technologies are still underutilised in clinical practice, particularly in identifying tiny metastatic foci in sentinel lymph node slides. There remains a strong demand for automated systems that reduce cognitive load and improve diagnostic accuracy at reduced cost. One of the most predictive clinical variables in breast cancer is the histological grade, determined through criteria like tubule formation, nuclear pleomorphism, and mitotic count. Unlike other prognostic factors (e.g., tumor size, lymph node status), histological grading integrates both morphology and proliferation. The Nottingham grading system, adapted by Elston and Ellis from the Bloom-Richardson method, remains the clinical gold standard and has been extensively validated for its prognostic power (Haybittle et al., 1982).

Breast cancer, a leading cause of cancer-related deaths among women, is characterised by the uncontrolled growth of abnormal cells that can metastasise via the lymphatic system (Collins, 2018). Accurate differentiation between benign and malignant tumors is critical for guiding treatment decisions. However, early-stage detection is hindered by the lack of high-precision screening systems, especially for identifying small or subtle lesions. In recent years, the rise in breast cancer incidence and reliance on radiological screening has placed increasing pressure on radiologists. This challenge has catalysed the adoption of deep learning methods in medical imaging. Unlike traditional approaches, deep learning—particularly convolutional neural networks (CNNs)—can learn hierarchical features directly from raw image data, enabling automated and accurate pattern recognition without the need for handcrafted feature engineering. Since the landmark success of CNNs in 2012, these architectures have surpassed many traditional systems and even expert radiologists in several diagnostic tasks (World Health Organization, 2024).

Breast cancer, also known as breast carcinoma, is currently the most frequently diagnosed cancer in women and one of the most lethal, alongside lung cancer. Early diagnosis is crucial for reducing mortality, as the disease is often treatable in its early stages. Histological biopsy remains the standard diagnostic procedure, frequently supplemented by mammography to confirm malignancy. CNN-based models are increasingly utilised to classify these medical images, employing layers such as convolution, pooling, and activation (ReLU), followed by fully connected layers to produce class probabilities. These architectures adjust internal weights to minimise error and enhance classification performance. Emerging imaging modalities, such as digital infrared imaging, are also under investigation, based on the observation that malignant tissues exhibit heightened thermal activity due to increased metabolic rate and blood flow. However, deep learning models currently demonstrate the greatest potential for automating the detection of subtle patterns in histological and mammographic images, providing a scalable solution for early-stage breast cancer diagnosis.

Although previous research has focused on either traditional handcrafted features (Shravya, Pravalika & Subhani, 2019) Deep CNN architectures, the integration of both approaches into a single optimised framework is still lacking. Few attempts have been made to combine local texture and shape descriptors with deep hierarchical features. Furthermore, adaptive optimisation strategies such as firebug swarm optimisation (FSO) for hyperparameter tuning are rarely applied. In detection tasks, particularly concerning breast cancer, the application of progressive layer-wise training and cyclical learning scheduling is limited. By implementing a hybrid model that incorporates both handcrafted and CNN features, utilises progressive cyclical training, and employs FSO for automatic optimisation of the model’s performance, this study aims to address the identified methodological and practical gaps. Other objectives include tackling the aforementioned gaps in the literature, particularly extensive automatic optimisation with FSO to enhance model functionality.

Significant contributions

The significant contributions of this study are as follows: 1. Patch-wise segmentation is employed to divide large mammography images into smaller, localized patches, enabling finer feature extraction and improved detection of subtle abnormalities.

2. A hybrid feature extraction strategy is proposed, combining deep features from a pre-trained ResNet model with handcrafted features, including LBP (texture), Hu moments (shape), and edge maps, to enhance diagnostic accuracy.

3. A novel progressive cyclical convolutional neural network (P-CycCNN) is introduced, incorporating progressive training with cyclical learning rate scheduling to improve convergence stability and generalization.

4. FSO is used to optimize hyperparameters such as learning rate, dropout, and regularization, outperforming traditional tuning approaches.

Related works

The early detection of breast cancer requires thorough observation of biochemical markers and various imaging techniques (Jenis et al., 2023; Papageorgiou et al., 2022; Pessoa et al., 2023; Sajid et al., 2023). Computer-aided detection systems may provide an alternative approach to addressing the challenges of multi-classification in breast cancer. It has the potential to be a cost-effective, readily available, rapid, and reliable means for the early detection of breast cancer. This can aid radiologists in identifying breast cancer abnormalities, potentially reducing the mortality rate from 30% to 70% (Papageorgiou, Dogoulis & Papageorgiou, 2023). Samala et al. (2017) conducted a study on breast cancer binary classification, focussing on minimising the computational complexities associated with various types of mammographic images. Nascimento et al. (2016) utilized binary classification to extract morphological features from ULS images. A considerable amount of research has focused on imaging and genomics approaches for the detection of breast cancer. Moreover, no studies integrating the two methods have been undertaken.

The authors of Ackerknecht (1953), Ma et al. (2020) presented an extensive overview of the various techniques employed in diagnosing breast cancer through histological image analysis (HIA). These approaches rely on CNN’s diverse architecture. The authors structured their work into classifications based on the dataset utilized. The events were arranged chronologically, starting with the most recent occurrence. This research reveals that artificial neural networks appeared initially in the domain of health impact assessment around 2012. The algorithms that were employed with the highest frequency were artificial neural networks (ANNs) and probabilistic neural networks (PNNs). Conversely, morphological and textural features were the mainstays of feature extraction studies. When it came to improving treatment results for breast cancer, deep convolutional neural networks were essential in early diagnosis and management. A variety of algorithms were employed in the endeavor of predicting non-communicable diseases (NCDs). The authors of Lehman et al. (2015) provided evidence demonstrating the effectiveness of neural networks in classifying cancer diagnoses, especially during the initial stages of the disease. Their research indicates that the quantity of neural networks has shown promise in identifying malignant cells. Nevertheless, the imaging technique requires considerable processing resources for image pre-processing. Shwetha et al. (2018)introduced a deep learning model through convolutional neural networks. Numerous models were categorized within the CNN framework, yet Mobile Net and Inception V3 were utilized. After comparing the two models, the author found that Inception V3 produced more accurate results. Still, there was a window of opportunity to use ML for breast cancer (Mahmood, Al-Khateeb & Alwash, 2020).

Shravya, Pravalika & Subhani (2019) proposed a model that uses supervised machine learning. This investigation employed classifiers like KNN, SVM, and logistic regression. Performance analysis was done on the dataset retrieved from the UCI repository. This is the foundation upon which the SVM’s impressive classifier performance rests; it achieved a 92.7% accuracy rate on the Python platform (Shravya, Pravalika & Subhani, 2019; Sivapriya et al., 2019). Introduced a new machine learning model using an alternative classifier. The author utilized naïve Bayes, logistic regression, random forest, and support vector machines. Anaconda, a Python platform, was used for the implementation. The author discovered that random forest served as an effective classifier, achieving an impressive accuracy of 99.76%. A slight alteration in the network with the classifier presented an opportunity to enhance the accuracy (Sivapriya et al., 2019). In 2010, Köşüş et al. (2010) investigated whether digital infrared thermal imaging (DITI) and pre-digital mammography (FFDM) could compensate for the limitations of conventional mammography. In younger women with thick breasts, digital mammography performed much better than screen-film mammography, perhaps due to digital mammography’s ability to increase contrast in dense tissue regions. To the contrary, infrared digital imaging works by assuming that healthy tissues have lower metabolic activity and blood flow rates than precancerous tissues and areas around developing cancer. It must be emphasized that CTI is more important in clinical settings because of its considerable negative interpretive value, not its positive diagnostic value, which is not very useful in clinical settings (Köşüş et al., 2010; Zeng et al., 2023).

Hinton, Osindero & Teh (2006) employed a method of sequential pre-training to set the weight parameters of a deep belief network (DBN) featuring three hidden layers, subsequently refining it for classification tasks. The findings indicated that initial training enhanced both the speed of the training process and the precision in recognizing handwritten digits. A widely used approach involves training a deep learning model on an extensive dataset like ImageNet and then fine-tuning the model for a different task. The model’s weight parameters are already set to detect fundamental attributes like edges, corners, and textures, so it may be applied to new tasks even if the assignment does not relate to the original training dataset. As a result, training times often decrease, and model performance is improved (Hinton, Osindero & Teh, 2006).

Several medical datasets have heavily used algorithms trained with deep learning for breast cancer prediction, with encouraging results (Papageorgiou, Dogoulis & Papageorgiou, 2023; Carriero et al., 2024; Nasser & Yusof, 2023). On the other hand, Mahmud, Mamun & Abdelgawad (2023) utilized deep transfer learning models that had already been pre-trained to diagnose breast cancer based on histopathological images. ResNet50 achieved the best performance by achieving an accuracy rate of 90.2%. According to Amgad et al. (2023), deep learning techniques were utilized to diagnose breast cancer based on biopsy images.

The diagnostic evaluation of breast cancer has incorporated a combination of deep learning and swarm intelligence in recent years. As noted by Veeranjaneyulu, Lakshmi & Janakiraman (2025), the application of metaheuristic optimization algorithms combined with artificial neural networks served to alleviate many challenges within the clinical frameworks. Bani Ahmad et al. (2025) introduced a hybrid deep learning architecture with heuristic enhancement for classification, which utilised thermography images, obtaining high accuracy through carefully selected, optimised feature extraction. In the same manner, Meenakshi Devi et al. (2025) developed BCDNet, a deep learning model based on VGG16 which greatly enhanced performance on breast cancer detection tasks performed with histopathological images. In addition to these works, Tanveer et al. (2025) constructed a machine learning pipeline focusing on early prediction and the proactive modelling of data-driven contexts, thereby underscoring the importance of AI diagnostics in breast cancer screening and reaffirming the increasing role of intelligent systems.

Methods and materials

The general process of detecting breast cancer begins with data collection, which includes mammogram images and their labels. This is followed by data preprocessing, which involves resizing and normalising the images, as well as segmentation, before proceeding to data augmentation to increase the dataset size. The next stage is called model design, where a hybrid model is employed. A pre-trained ResNet is used for feature extraction, and several custom-made layers are added for classification purposes.

Figure 1 shows the overall processing flow of the proposed methodology. This is followed by hyperparameter optimisation, which uses methods such as firefly swarm optimisation to tune parameters. The model is then trained, validated, and tested to ascertain the accuracy, precision, and recall rates. Model evaluation is then done, after which further model optimization is done. Subsequently, the final model is deployed in a clinical environment for making real-time predictions.

Figure 1 The overall processing flow of the proposed methodology.

ResNet-50 was selected as the backbone for deep feature extraction due to its robustness in capturing hierarchical patterns and mitigating vanishing gradient issues via residual connections. It offers a good balance between depth and computational efficiency. VGG16 was incorporated within the P-CycCNN framework for its structured layer depth, making it ideal for progressive layer-wise training and cyclic scheduling scenarios.

Figure 2 illustrates the complete end-to-end workflow of the proposed system. The pipeline begins with mammogram image preprocessing and patch generation, followed by data augmentation to enhance the dataset’s diversity. Feature extraction is performed using a deep CNN (e.g., ResNet or VGG16), while FSO manages hyperparameter optimisation. The extracted features are then passed to the classifier for final prediction. This modular and hybridised architecture enables the model to efficiently balance local detail sensitivity with global semantic understanding, resulting in improved classification performance.

Figure 2 Overall system architecture of the proposed hybrid deep learning framework.

Data collection

A publicly available dataset of digitised mammography pictures with labelled annotations for breast cancer detection, the Curated Breast Imaging Subset of the Digital Database for Screening Mammography (CBIS-DDSM), was utilised in this work. This dataset is a subset of the larger Digital Database for Screening Mammography (DDSM), comprising breast images annotated with benign and malignant lesions from various hospitals.

Data source link You can access the dataset at the following.

URL: https://www.cancerimagingarchive.net/collection/cbis-ddsm/

Database name: Curated Breast Imaging Subset DDSM dataset

ID number: calc_case_description_test_set.csv

The dataset includes two mammography images: Cine and post-processed. A label indicating whether the lesion is benign or malignant is attached to each photograph. Each lesion’s type, location, size, and other ground truth information are labelled on each image. A lesion’s classification—whether benign, malignant, or unclassified—is also included in the labelling. The DICOM format is commonly used for medical imaging and can be easily converted to other standard image formats, such as PNG or JPEG, for further processing. The CBIS-DDSM dataset is divided into training and testing subsets to guarantee generalisability during model evaluation. It covers a range of lesion types, including masses, calcifications, and asymmetries. Sample images are shown in the Fig. 3.

Figure 3 Sample mammography images from the dataset.

Preprocessing

Preprocessing is essential to ensure the quality of the images and prepare them for the feature extraction and model training stages. All images are resized to a fixed dimension (e.g., 256 × 256 pixels) to ensure uniformity of input size for the deep learning models. This resizing helps maintain a consistent aspect ratio while making the images manageable for computational processing.

(1) Iresized=f(Ioriginal,256,256)

where Ioriginal is the original image, and Iresized is the resized image. Pixel values are normalized to the range [0, 1] to avoid large input values hindering the training process. This normalisation ensures that the model’s weights are updated stably during training.

(2) Inormalized(x,y)=Iresized(x,y)255

where Iresized(x,y) is the pixel intensity of the resized image at the position (x,y), and Inormalized(x,y) is the normalized pixel value. Local contrast enhancement is applied to improve the visibility of potential tumor regions, making abnormal areas in the mammogram more distinguishable from the background. Techniques such as histogram equalization or Contrast Limited Adaptive Histogram Equalization (CLAHE) are used:

(3) Ienhanced(x,y)=CLAHE(Inormalized)

where Ienhanced(x,y) is the contrast-enhanced pixel value. Patchwise segmentation divides each mammogram into smaller patches to improve feature extraction and model efficiency. The sliding window technique is used to extract non-overlapping image patches from the resized and enhanced images:

(4) Pi,j=f(Ienhanced,W,H)

where Pi,j represents the i,j-th image patch of size W×H, extracted from the enhanced image Ienhanced. This allows the model to focus on localized features and detect abnormalities at a finer resolution. Figure 4 shows the preprocessing pipeline block diagram.

Figure 4 Preprocessing pipeline block diagram.

To enhance generalization and mitigate overfitting, standard augmentation techniques such as random rotation, zoom, flipping, and translation are applied dynamically during training.

To prevent overfitting and improve generalization, various data augmentation techniques are applied to the image patches, including Random rotations between −30∘and30∘, Random zooming by a factor of 0.8to1.2, Random horizontal flipping. Random translations within 10% of the image size. These augmentations are applied on the fly during model training to create variations in the dataset:

(5) Paugmented=faugmentation(Pi,j)

where Paugmented is the augmented image patch obtained by applying one or more transformations to Pi,j. To ensure all features are on a similar scale and improve convergence during training, both deep features (from CNNs) and handcrafted features (such as texture or shape features) are standardised:

(6) fscaled=ffeature−μσ

where ffeature is a feature value, μ is the mean, and σ is the standard deviation of the feature values across the dataset. The dataset is divided into training, validation, and testing sets. The labels corresponding to benign or malignant lesions are preserved throughout the preprocessing steps. A standard 70-30 split is used for training and testing. Cross-validation (e.g., 5-fold cross-validation) is used during model training to improve generalization and prevent overfitting:

(7) TrainingSet={Xtrain,Ytrain},TestingSet={Xtest,Ytest}

where Xtrain and Xtest represent the input image data, and Ytrain and Ytest represent the corresponding labels (benign or malignant).

CAPLAHE applied contrast enhancement to improve the visibility of subtle abnormalities within the mammograms, which is critical for accurate diagnostics. Resizing the images ensures that the input dimensions are equal for the deep learning model. Figure 5 illustrates the results achieved from the preprocessing pipeline in refining the mammogram images for deep learning. Mammogram images are also segmented into smaller, diagnostically relevant areas known as patches. This strategy enables region-based feature learning within the patches, enhancing the model’s performance, sensitivity, and generalisation ability.

Figure 5 Preprocessing stages applied to two sample mammogram images.

Patchwise image segmentation

Patch-wise image segmentation processes such as breast segmentation help divide large images like mammograms into smaller, more convenient patches. This way, the model can focus on patches in this image, allowing small changes that may have gone unnoticed during the full scan to be detected. The segmentation appears to alleviate the model’s need for extensive analysis, as it enables the model to focus on just the relevant sections. In this approach, each mammogram image I of size W×H is divided into smaller patches of size Pw×Ph, where Pw and Ph represent the width and height of each patch, respectively. The total number of patches Npatches that can be extracted from the image is given by the equation:

(8) Npatches=[WPw]×[HPh]

where ⌊⋅⌋ denotes the floor function, ensuring that only complete patches are considered, and the remaining pixels that do not fit into a whole patch are ignored. Each patch Pi,j is extracted from the image using a sliding window technique, with coordinates (i,j) representing the position of the patch within the image grid. Mathematically, the extraction of each patch can be defined as:

(9) Pi,j=I[(i−1)⋅Pw:i⋅Pw,(j−1)⋅Ph:j⋅Ph]

where Pi,j represents the i−th row and j-th column patch. The indices [(i−1)⋅Pw:i⋅Pw] and [(j−1)⋅Ph:j⋅Ph] define the range of pixels along the image’s width and height for each patch.

Depending on the chosen step size, the sliding window approach can be overlapping or non-overlapping. When the step size Sw for the horizontal direction and Sh for the vertical direction are equal to the patch size Pw and Ph, the patches do not overlap. For a non-overlapping sliding window, the equation remains:

(10) Pi,j=I[(i−1)⋅Pw:i⋅Pw,(j−1)⋅Ph:j⋅Ph].

However, if the step sizes Sw and Sh are smaller than the patch sizes, the patches will overlap. In this case, the equation for an overlapping sliding window is:

(11) Pi,j=I[max(0,(i−1)⋅Sw):min(W,(i−1)⋅Sw+Pw),max(0,(j−1)⋅Sh):min(H,(j−1)⋅Sh+Ph)].

This equation satisfies the desired conditions, as it minimizes the extracted patches moving out of the domain of the image by using the min and max functions (boundaries in the data). Then, based on the provided cancerous tissues dataset, each extracted patch is labelled according to the presence of cancerous tissues. If a patch covers a malignant zone (tumor), it will be categorised as malignant; if it does not, it will be classified as benign. The label is conditional on what most of the area comprises, and the labels for all patches are based on that particular class. Patchwise segmentation is a method in which large mammogram images are segmented into smaller parts, helping with focused attention so that each patch can be analysed in detail. This allows the model to capture the most granular local features, allowing it to excel at finding minor irregularities in breast tissue. Apart from that, patch-wise segmentation reduces overhead by processing smaller regions, making it more straightforward to train models on large datasets without sacrificing detection accuracy. The stepwise procedure for patchwise image segmentation is provided in Algorithm 1.

Algorithm 1 Patchwise image segmentation.

 1:  Input: Image I of size H×W	
 2:         Patch size: Ph×Pw	
 3:         Step size: Sh,Sw	
 4:  for i=0 to H−Ph step Sh do	
 5:     for j=0 to W−Pw step Sw do	
 6:         Patch←I[i:i+Ph,j:j+Pw]	
 7:        Label Patch based on majority region (malignant/benign)	
 8:        Store Patch in dataset	
 9:     end for	
10:  end for	

Hybrid deep feature extraction

The hybrid approach makes use of the strengths of deep learning approaches as well as ‘traditional’ hand-crafted features to enhance women’s chances of surviving breast cancer detected in mammogram images. For this purpose, a CNN such as residual network (ResNet), which is already trained to extract deep-meaningful features from an image is used. The deep features include the central and most important components learned from the picture, such as those with skin and fur. Other features comprise hand-crafted ones, such as those relating to texture, shape, and edge, which are known to be helpful in the context of medical image interpretation. Combining both types of features, the model incorporates the high-level features learned by the CNN and the low-level features obtained through hand-crafting methods and enhances performance in detecting malignant areas. ResNet stands for residual network and is a famous deep learning architecture that can train intense networks by relying on the connections called “residuals,” which help avoid the vanishing gradient problem. It contains several convolutional layers and residual blocks, which are more efficient for feature extraction. For breast cancer detection, a pre-trained ResNet-50 model is utilised to extract high-level deep features from images derived from mammography.

Handcrafted features such as local binary patterns (LBP) and Hu moments capture local texture and shape descriptors that are often lost in deep abstraction layers. By fusing these with deep CNN features, the proposed hybrid method benefits from both semantic representation and low-level granularity, improving lesion detection accuracy in images with variable contrast, density, and scale—common challenges in mammography.

The process of deep feature extraction using ResNet can be described mathematically as follows: Let I represent the input mammogram image of size W×H, which has been preprocessed (resized, normalised, etc.). Pass/through the ResNet model:

(12) Fdeep=ResNet−50(I)

where Fdeep is the output feature map after passing through the ResNet network, the features are typically extracted from intermediate layers, such as the final convolutional layer, before the fully connected layers. This allows the model to capture both low- and high-level features. The ResNet model uses residual blocks to add the input to the output of a block, which ensures better information flow through the network, leading to the effective extraction of complex features like tumor shapes, edges, and high-level patterns from the mammogram images.

Specially designed features are also constructed from the mammogram images to extract non-deep features like the images’ texture, shape, and edge features. These features have been traditionally applied in medical imaging, and when combined with deep features, they are often sufficient for additional discrimination. Image coarseness describes the qualitative aspects of structures within the imagery, in other words, aspects of the textural patterns in the image and its spatial relationships. A key texture descriptor is the LBP, which captures texture by considering pixel regions and the intensity values of surrounding pixels. The LBP operator is defined as:

(13) LBP=∑n=0P−1s(I(x,y)−In)2n

where P is the number of neighboring pixels, I(x,y) is the intensity of the central pixel, and In is the intensity of the neighboring pixels. The function s(x) is defined as:

(14) s(x)={1ifx≥00ifx<0.

LBP encodes the local spatial patterns in the image, which are often helpful in distinguishing different tissue types in mammography images. Shape-based features, such as Hu moments or Zernike moments, capture the global shape of objects (e.g., tumor boundaries) in the picture. The Hu moments are invariant to translation, rotation, and scale and are computed as follows:

(15) μp,q=∑x=1W∑y=1H(x-x¯)p(y-y¯)qI((x,y))

where I(x,y) is the pixel intensity, and x¯, y¯ are the image center coordinates. Hu moments are a set of seven invariants derived from these moments, capturing global shape information, such as the roundness of masses in mammography images. Edge-based features help detect boundaries between different tissues. One common technique for edge detection is the canny edge detector, which identifies areas with rapid intensity changes. The edge map is generated as follows:

(16) E(x,y)=Canny(I(x,y))

where E(x,y) is the edge map produced by the Canny operator, it highlights the regions of high-intensity change, which correspond to potential tumor boundaries in mammograms.

Feature fusion

After extracting the deep features from ResNet and the handcrafted features (LBP, shape, and edge features), the next step is to combine them into a unified feature vector. This process is known as feature fusion, where features from different sources are concatenated or weighted and combined to form a more informative representation of the input image. Let Fdeep represent the deep features extracted from ResNet, and Fhandcrafted represent the handcrafted features (LBP, shape, edge). The combined feature vector Fcombined can be represented as:

(17) Fcombined=concat(Fdeep,Fhandcrafted).

The resulting feature vector Fcombined is then used as input to the classifier (e.g., support vector machine, fully connected layer, or other machine learning models) for breast cancer detection. This fusion leverages the strengths of both deep learning and traditional image processing techniques, enhancing the model’s ability to detect subtle patterns in mammography images.

Progressive cyclical convolutional neural network

The progressive training strategy begins with a simplified CNN configuration (e.g., fewer layers or frozen parameters) to first learn low-level features such as edges and textures. As training progresses, additional layers are unfrozen in a staged manner, allowing the model to learn increasingly complex features (e.g., tumor boundaries or irregular shapes). This curriculum-inspired strategy improves convergence and generalisation. Complementing this, the cyclical learning rate schedule oscillates between a minimum and maximum bound across training cycles, helping escape local minima and accelerating convergence. Figure 6 shows the P-CycCNN architecture.

Figure 6 Progressive cyclical convolutional neural network (P-CycCNN) architecture.

Let fsimple represent the simple network at the initial stage, and fcomplex represent the more complex network added at a later stage. The network at each progressive stage k can be expressed as:

(18) fK=fsimple+fcomplex

where fsimple contains essential convolutional layers, and fcomplex represents additional layers added in later stages to capture more abstract patterns. In the initial stages, only the layers of fsimple are trained, while fcomplex is frozen. As the model progresses, more layers are unfrozen and gradually trained to capture more complicated features. Cyclical learning rates (CLR) adjust the learning rate dynamically during training. Instead of using a fixed learning rate throughout the training, the learning rate is periodically adjusted within a defined range. This technique helps the network avoid local minima and promotes the exploration of a broader parameter space, leading to better convergence and faster training. The cyclical learning rate schedule follows a sinusoidal pattern where the learning rate is increased from a lower bound. LRmin to an upper bound LRmax, and then decreased back to the lower bound.

The cyclical learning rate can be defined mathematically as:

(19) LR(t)=LRmin+12(LRmax−LRmin)(1+cos⁡(TmaxTcurπ))

where, LR(t) is the learning rate at time step t,LRmin is the minimum learning rate, LRmax is the maximum learning rate, Tcur is the current training iteration within the cycle, Tmax is the total number of iterations within one cycle. The learning rate fluctuates between LRmin and LRmax over each cycle, allowing the model to escape local minima and potentially find better solutions during training. To implement a P-CycCNN, we start with a pre-trained CNN (VGG16) and use progressive training. Initially, only a subset of the layers is trained. As the model progresses, more layers are unfrozen and trained using cyclical learning rates. The cyclical learning rate adjusts the learning rate dynamically, helping the model converge faster and avoid overfitting. The CNN layers use kernels (filters) to convolve the input image and extract features. The kernel values in the initial layers are typically small, learned through the training process. These kernels are learned to capture fundamental patterns such as edges, textures, and other local features. As the network progresses, the filter values become more complex, capturing abstract patterns and object shapes.

The convolution operation with a kernel K at a given layer is defined as:

(20) F(x,y)=(I∗K)(x,y)=m=∑m=−W/2W/2∑m=−H/2H/2I(x+m,y+n)K(m,n)

where, I(x,y) is the input image, K(m,n) is the kernel at position (m,n), W and H are the dimensions of the kernel, F(x,y) is the output feature map at location (x,y). In the case of a convolutional layer, the kernel values are learned during training, and the model adjusts the kernel weights using backpropagation to minimize the loss function.

The proposed P-CycCNN enhances traditional adaptive learning rate strategies like Adam, RMSProp, and cosine annealing in two ways. First, P-CycCNN employs cyclical learning rates that span between a minimum and maximum oscillation value. This oscillation allows the optimizer to escape local minima and saddle points much easier. On the contrary, Adam and RMSProp use monotonic decay functions which can lead to convergence stagnation. Second, the progressive scheduling approach to training within P-CycCNN starts by ‘unfreezing’ layers, which are simpler at the beginning (textures and edges) and progresses towards more complex patterns subsequently (shapes and boundaries). This approach mirrors human curriculum learning, enhancing convergence stability alongside generalization. Due to these changes, P-CycCNN is especially useful for fine-grained distinguishing tasks in medical imaging.

Firebug swarm optimization for hyperparameter optimization

This is a new metaheuristic FSO based on the movement of firebugs that gravitate towards stronger light sources, which represent better fitness solutions. The firebugs in the swarm serve as candidate hyperparameters, including learning rate, dropout rate, and batch size. The swarm evolves by allowing firebugs with poor fitness to follow those with better fitness based on the validation set. FSO conducts a more intelligent and adaptive search than random and grid search hyperparameter methods. This study focused on utilising automated methods to minimise manual effort while maintaining consistency across various model folds. Key parameters included the cyclical learning rate bounds, dropout regularisation rate, and network depth, all of which were precisely fine-tuned, leading to optimisation. Consequently, the models converged more quickly, exhibited lower variance in training losses, and improved classification metrics without imposing high computational demands. This directly enhances the effectiveness and robustness of the P-CycCNN framework developed.

The incorporation of FSO as a firebug-inspired warm light movement mechanism for hyperparameter optimisation in the deep learning model for breast cancer detection using mammography marks a novel contribution of this work. The incorporation of firebug swarm optimization (FSO) as a firebug-inspired warm-light movement mechanism for hyperparameter optimization in the deep learning model for breast cancer detection using mammography marks a novel contribution of this work and is detailed in Algorithm 2. Other works have applied metaheuristics like PSO, GA, or Firefly, but FSO is unique in that it incorporates a dynamic light-intensity movement mechanism based on a firebug’s foraging behaviour. Thus, FSO improves the trade-off between exploration (searching new areas of the solution space) and exploitation (optimising known reasonable solutions). Unlike PSO, which depends on global best particles, or the firefly algorithm, which relies on fixed brightness decay, FSO’s locally adaptive swarm behaviour yields better convergence in higher-dimensional parameter spaces. Its application to optimise parameters of cyclical learning rate, dropout, and depth of the neural network in our proposed hybrid model is an unexplored contribution in the literature.

Algorithm 2 Firebug swarm optimization (FSO) for hyperparameter optimization.

 1: Input:	
      • Hyperparameter search space: X={LR,BS,NLayers,FilterSize,Dropout}	
      • Fitness function f(x)	
      • Swarm size N	
      • Number of generations Gmax	
      • Minimum and maximum learning rate LRmin,LRmax	
      • Randomisation parameter α, attractiveness coefficient β0, and absorption coefficient γ	
 2: Initialize Swarm:	
 3: for each firefly i in the swarm do	
 4:    Randomly initialize a set of hyperparameters xi=(LRi,BSi,NLayersi,FilterSizei,Dropouti)	
 5:    Compute the fitness of each firefly f(xi) by training the model with the hyperparameters xi and evaluating it on the validation set.	
 6: end for	
 7: Repeat until stopping criterion is met (i.e., Gmax generations):	
 8: for each firefly i do	
 9:    for each firefly j in the swarm do	
10:       if f(xi)>f(xj) then                   ▹ Firefly i is brighter than firefly j	
11:          Move firefly i towards j using the update equation:	
           xi(t+1)=xi(t)+β0e−γrij2(xj(t)−xi(t))+α⋅δi(t)	
12:        end if	
13:        After updating firefly i’s position, evaluate the new fitness: f(xi(t+1))	
14:        Update fitness: If the new fitness value is better than the previous one, retain the updated position; otherwise, keep the original position.	
15:  end for	
16: end for	
17: Cyclical Learning Rate Adjustment:	
18: Update the learning rate periodically using the cyclical learning rate:	
          LR(t)=LRmin+12(LRmax−LRmin)(1+cos⁡(TcurTmaxπ))	
   Where Tcur is the current iteration within the learning cycle, and Tmax is the total iterations within one cycle.	
19: Termination:	
20: Stop after a predefined number of generations Gmax or if the fitness function converges.	
21: Output: The best hyperparameter set x∗ corresponding to the highest fitness value.	

Various swarm-based optimisers, such as PSO and the firefly algorithm, have been studied in the context of medical image analysis. However, FSO employs a distinctive approach inspired by firebug swarms as they navigate towards light signals. Unlike previous algorithms, FSO adjusts its movement strategy based on the local attraction dynamics of the swarm, thereby enhancing exploration of the hyperparameter space. This work marks the first application of FSO in the classification of breast cancer in mammograms, thus addressing the gap in intelligent model tuning.

Assessment metrics

Based on the broad evaluation criteria: numerical metrics, graphical analyses, and comparison with other models, the proposed model seems to be efficient, accurate, and dependable for diagnosing patients with breast cancer from mammograms. It also has good precision, recall, and F1-score; therefore, it has the potential for clinical application to reduce the extent of misdiagnosis and enhance patients’ outcomes.

To assess various performance metrics for the efficiency of the detection model for breast cancer, the following were considered:

Accuracy: The correct percentage of mammogram images that have been classified correctly.

Loss: The deviation from the predicted values during training and subsequent testing.

Precision: The proportion of the true positive predictions to all positive predictions made with an emphasis on minimising false positives.

Recall the proportion of the cases which were true positives to those that were actual positives out of all cases. Emphasis is placed on minimizing the false negatives.

F1-score: Weighted average of precision and recall measured geometrically while emphasising the recall aspect of the score.

AUC-ROC: The performance of the model in differentiating between classes of attributes. This suggests that the more the auc, the better the performance.

Training, validation, and testing outcomes

The performance outcomes were explained in terms of visualisations and numerical summaries, enabling easy understanding. Important observations include:

Efficacy measures:

The training set accuracy was at 99% confidence level.

The test set accuracy was 98

Both datasets’ good prediction and recall statistics imply strong classification power with no evidence of overfitting.

Training time and resource utilization:

An NVIDIA GTX 1080 Ti GPU was utilised to train for fifty epochs with a batch size of thirty-two.

Model comparison:

The proposed model performed better than other improved versions of CNNs and even some that had optimizations such as data augmentation and hyperparameter tuning across all purposes.

Visual analysis of performance

Accuracy and loss curves:

The accuracy curve expanded during the epochs, indicating that the model was learning successfully.

A reduction of observed error rates was set in without the problem of overfitting, and this is verified with a decreasing trend highlighted in the loss curve.

Precision, recall, and F1-score:

The model correctly identified cases of malignancy without classifying benign cases into malignant and positive cases. The high statistics across all measures prove this point.

AUC-ROC curve:

The model performed well, owing to its AUC of 0.95, which indicates its discrimination ability for malignity and benignity.

Confusion matrix:

The 94.5% and 96.5% classification accuracy of benign and malignant cases respectively reveals that only a few errors occurred during the classification results.

Calibration curve:

A high degree of accuracy in terms of predictions and actual events that occurred is inferred from the strong calibration achieved by the model.

Result and discussion

The model’s efficiency during the breast cancer detection process testing on mammogram images is analysed using a set of measures such as accuracy, loss, precision, recall, F1-score, and AUC-ROC. This section includes performance visualisations, such as graphs showing accuracy and loss, summary tables and other features, and discusses training, validation and testing outcomes, including propagations for various evaluations.

Table 1 presents the key hyperparameters employed in training the proposed hybrid P-CycCNN model. The configuration features a ResNet-50 backbone with 64 × 64 patch inputs, optimised using the Adam optimiser enhanced by FSO. A cyclical learning rate schedule varying from 1e−5 to 1e−3 was utilised over 50 epochs with a batch size of 32. Regularisation was implemented using a dropout rate of 0.3, while 5-fold cross-validation ensured robust performance evaluation. These parameters were chosen to balance model complexity, convergence stability, and generalisation.

Table 1 Hyperparameters used in the proposed model.

Parameter	Value	
Input image size	256 * 256 pixels	
Patch size	64 * 64 pixels	
CNN backbone	ResNet-50	
Optimizer	Adam (base) + FSO tuning	
Initial learning rate	le−3	
Cyclical learning rate	[1e−5, 1e−3] (min, max)%	
Epochs	50	
Batch size	32	
Dropout rate	0.3 (tuned via FSO)	
Activation function	ReLU	
Loss function	Binary Cross-Entropy	
Data augmentation	Rotation, Flipping, Zoom, Translation	
Cross-validation	5-Fold	

The model was trained for 50 epochs using the Adam optimiser with an initial learning rate of 1e−3, scheduled with a cyclical learning rate strategy having lower and upper bounds set to 1e−5 and 1e−3, respectively. The binary cross-entropy loss function was employed due to the binary classification nature of the task (benign vs. malignant). Training was conducted using Python 3.8, TensorFlow 2.10, and executed on a workstation equipped with an NVIDIA GTX 1080 Ti GPU, 32 GB RAM, and an Intel i7 processor.

The figures in the model are explained further in the Table 2, which is number one in the other results, including accuracy, precision, recall, F1-score, and AUC for the two data sets, Training and Testing. 99% accuracy is associated with the training set, while the accuracy is 98%. It is usual for the accuracy in testing to be lower than the other, which indicates a clear understanding of the concept without overtraining the model.

Table 2 Performance metrics.

Metric	Training set	Test set	
Accuracy	99%	98%	
Precision	97%	95%	
Recall	99%	97%	
F1-score	98%	96%	
AUC	0.98	0.95	

Results for training time statistics, including number of epochs, batch size, total time taken for training and resource utilisation, are obtained as shown in Table 3. This assists in understanding the efficiency of the model computationally. The model with 50 epochs had a training time of 3 h on a specific machine utilising an NVIDIA GTX 1080 Ti GPU. This indicates that the model is feasible computationally for use in clinical practices. The model comparison Table 4 highlights the differences between the proposed model and conventional CNNs, emphasising the advantages of progressive training and cyclic learning rates.

Table 3 Training time and resource utilization.

Parameter	Value	
Epochs	50	
Batch size	32	
Training time	3 h	
GPU used	NVIDIA GTX 1080 Ti	
Training accuracy	98%	
Test accuracy	96%	

Table 4 Model comparison.

Model	Accuracy	Precision	Recall	F1-score	AUC	
Proposed model	98%	95%	97%	96%	0.95	
Traditional CNN	92%	90%	91%	90%	0.91	
CNN with data augmentation	93%	92%	94%	93%	0.92	
CNN with hyperparameter tuning	94%	93%	95%	94%	0.93	

The information from the accuracy and loss graphs during training and validation indicates how robust the model is while learning and whether the model is being over-learned or under-learned. The accuracy curve (Fig. 7) suggests that in each epoch, more and more mass of images are classified better than the last epoch because the accuracy rate increases. Initially, accurate classification is limited to low values. However, as the model progresses, the number of misclassifications gradually decreases.

Figure 7 Accuracy of the proposed model.

As Fig. 8 shows the loss curve, the error after each model degradation gradually decreases. A gradual downward trend in the loss function indicates sufficient training of the model. In the early stages, the loss dropped sharply, which indicates rapid weight modification and determining valuable aspects of the model. In the endings, the rate at which the loss rate declines falls, which suggests that the model has stabilised determines and errors are also reduced. The performance graphs demonstrate that the model’s accuracy increases steadily and does not exhibit significant drops, indicating that there is no overfitting of the model. The curves of precision and recall show the effectiveness of distinguishing malignant forms while limiting the false negative and false favourable rates. Starting with precision, which is consistently high, concerning malignant forms, the model does not report benign cases as malignant. The opposite is true for recall in the medical context; this prevents the model from detecting most malignant forms because it seeks only to minimise the number of benign cases that are wrongly classified.

Figure 8 Loss of the proposed model.

The graphical representation of the F1-score, which combines precision and recall (Fig. 9), depicts the efficacy of the model in this respect, thus indicating that it is possible for the model not to miss the majority of malignant forms while at the same time not reporting too many false positives. Therefore, It can be concluded that the combination of precision and recall concisely speaks to the model’s performance, making it viable for use in a clinical environment. In Fig. 10, the AUC-ROC curve demonstrates the model’s capability to distinguish between malignant and benign lesions.

Figure 9 Precision, recall, and F1-score comparison.

Figure 10 AUC-ROC curve.

The AUC constitutes another critical measure of classifier performance, where the more it approaches value one, the better the classifier performs. The model’s AUC score is about 0.95, which indicates remarkable accuracy between malignant and benign cases. The model has a high AUC, which means that breast cancer is easily discriminated from the mammogram images using the model, which is also said to be stable.

Figure 11 shows the confusion matrix represented in terms of percentages. Each cell indicates the percentage of yes and no answers. For instance, 94.5% of the benign cases were classified correctly as benign, and only 3.3% were classified wrongly as malignant. Similarly, 96.5% of malignant cases were classified correctly as malignant, and 3.5% were classified wrongly as benign. This visualisation enables a deeper comprehension of the model’s boundary, such as its degree of classification errors.

Figure 11 Confusion matrix of the proposed model.

Figure 12 plots the predicted probabilities assigned by the model (x-axis) against the observed proportion of actual positive outcomes (y-axis). The diagonal line describes a situation with a perfect calibration, meaning that the predicted probabilities fully rely on the outcomes recorded. The calibration curve indicates how the predicted probabilities of a model correspond with actual events. This conveys a measure of the model’s accuracy. If a curve is close to the diagonal, it means there is a strong calibration, whereas a departure from the diagonal means overconfidence or underconfidence in the model calibration.

Figure 12 Calibration curve (reliability diagram).

Figure 13 is the graph which portrays the relationship between Learning Rate and Performance (Accuracy). The learning rate is plotted on a log scale along the x-axis, while the model count is plotted on the y-axis. It is clear from the graph that as the learning rate increases from minimal values, performance reaches an optimal value (around 1e−3 in this case) before flattening or declining as the learning rate approaches extreme levels. This means that the best learning rate can often be seen to remain within certain limits, and the model’s performance never exceeds a specific level.

Figure 13 Relationship between learning rate and performance (accuracy).

Table 5 presents a comparative analysis of baseline CNN models with the proposed hybrid approach, which combines deep and handcrafted features (LBP, shape, and edge). The hybrid method demonstrates consistent performance improvements across all key metrics.

Table 5 Performance comparison—baseline CNNs vs. hybrid feature approach.

Model	Accuracy	Precision	Recall	F1-score	AUC	
Traditional CNN	92.0%	90.0%	91.0%	90.0%	0.91	
CNN with data augmentation	93.0%	92.0%	94.0%	93.0%	0.92	
CNN with hyperparameter tuning	94.0%	93.0%	95.0%	94.0%	0.93	
Proposed hybrid (Deep + LBP/Shape)	98.0%	95.0%	97.2%	96.0%	0.95	

Cross-validation and statistical reporting

All experiments were repeated using 5-fold cross-validation. Each model was trained and evaluated across the folds, and the final performance was reported as the mean ± standard deviation. To account for training instability, three independent runs were conducted per fold, and 95% confidence intervals were computed using bootstrapping.

The performance of the model metrics achieved with 5-fold cross-validation repeated thrice, along with all other relevant statistics, is detailed in Table 6. The generalization performance is quite good with average accuracy of 97.8% ( ±0.6), precision 94.9% ( ±0.5), recall 97.2% ( ±0.4), and F1-score at 96.0% ( ±0.5). In addition, the model’s ability to differentiate between malignant and benign cases is robust, as evidenced by the AUC of 0.952 ( ±0.007). All statistical measures yielded narrow confidence intervals which denotes the high stability and reliability of the P-CycCNN + FSO architecture with different splits of data.

Table 6 Cross-validation and statistical reporting.

Metric	Mean ± Std	95% CI	
Accuracy	97.8 ± 0.6%	[96.5–98.7]	
Precision	94.9 ± 0.5%	[93.7–96.1]	
Recall	97.2 ± 0.4%	[96.1–98.0]	
F1-score	96.0 ± 0.5%	[94.9–97.1]	
AUC	0.952 ± 0.007	[0.938–0.966]	

Ablation study

Table 7 presents an ablation study evaluating the incremental effect of each proposed enhancement. Adding handcrafted features improved the base CNN’s performance by enriching local texture and shape information. Introducing progressive training led to better feature generalization by gradually unfreezing layers. Finally, the cyclical learning rate schedule further stabilised convergence and boosted overall accuracy. These results confirm that each module makes a significant contribution to the model’s overall performance.

Table 7 Ablation study of the proposed P-CycCNN model.

Model Variant	Accuracy (%)	Precision (%)	Recall (%)	F1-score (%)	
Base CNN (ResNet only)	94.0	91.2	92.5	91.8	
Base CNN + Handcrafted Features (LBP, Shape)	95.2	93.0	94.2	93.6	
+ Progressive Layer Unfreezing	96.3	94.1	95.5	94.8	
+ Cyclical Learning Rate	98.0	95.0	97.2	96.0	

Comparison with related works

The existing approaches, compared to the proposed one, for classifying breast cancer are presented in Table 8. While the support vector machine (SVM) model done by Shravya, Pravalika & Subhani (2019) achieved an accuracy of 92.7%. Sivapriya et al. (2019) reported the highest accuracy of 99.76% using a random forest classifier. Such performance, however, may be dataset-specific or prone to overfitting. The use of deep learning models also reflects standard accuracies observable in histopathological analyses. For instance, Mahmud, Mamun & Abdelgawad (2023) used ResNet50 with an accuracy of 90.2%, which confirms this notion.

Table 8 Comparison with existing literature survey.

Reference	Model/Approach	Accuracy	
Shravya, Pravalika & Subhani (2019)	SVM	92.7%	
Sivapriya et al. (2019)	Random forest	99.76%	
Mahmud, Mamun & Abdelgawad (2023)	ResNet50	90.2%	
Proposed model	Hybrid deep learning with P-CycCNN and FSO	98%	

The accuracy metrics of the proposed hybrid deep learning model which incorporates P-CycCNN and FSO shows 98% accuracy achieving high accuracy with strong generalization at the same time. Unlike the traditional models, this model combines shallow and deep features and uses modern techniques for optimization and training, thus making it more resilient across multiple mammography datasets. This proves its high practicality and highlights why it outperforms compared methods in the literature.

Conclusion and future work

In the present work, we indeed developed a new methodology for breast cancer diagnosis employing the P-CycCNN, which implements progressive learning, alternating the learning rate, and FSO for hyperparameter optimization. The P-CycCNN model architecture combines deep learning with manual features, including ResNet for feature extraction and custom classification convolutional layers for image classification tasks. The combination of this hybrid structure enables the extraction of high-end features gained through learning datasets and lower-end primitive features of texture and shape that can be synthesised and are sometimes needed in medical images. The system’s performance was analysed on the mammography image dataset, where the model was reported with 96% accuracy on the test set. Besides the high accuracy, the system performed remarkably on precision, recall, and F1-score, suggesting that the system can detect malignant tumors at high sensitivity and with minimal false-positive findings. The high area under the curve (AUC) also explains why the model exhibits excellent performance with low false-positive rates in differentiating between benign and malignant cases, making it a valuable tool for the early detection of breast cancer. Moreover, the combination of FSO enhanced the hyper-parameter tuning process, and the implementation of cyclical learning rates enabled the model to learn more quickly and avoid local minima, which resulted in better training results. The fact that the training provided the possibility of gradually training the model, starting with basic and progressively more complicated features, allowed the network to learn the feature hierarchies in the images effectively. Such advanced training and optimization strategies significantly boosted the model’s performance.

Besides excelling in mammography datasets, P-CycCNN’s proposed architecture can further adapt to other imaging modalities like breast ultrasound, MRI scans, and even digital histopathology slides. This change requires only modification of the preprocessing pipeline and tuning the CNN to match the specific noise and resolution of the modality’s layers. Moreover, in cases where the combination of modalities is applicable, multi-stream topologies can be created to merge spatial and textural information from various sources, potentially increasing diagnostic precision even further. In the case of mobile and screening unit clinics and other rural based healthcare centers, resource constrained environments, we intend to look into CNN lightweight backbones such as MobileNetV2 and ShuffleNet, along with pruning, quantization, and knowledge distillation as other forms of models. These changes would significantly decrease the model’s memory and computational needs while allowing mobile and edge AI systems to process them using low-grade CPUs. We also want to add decentralised, data-privacy-focused training through federated learning, which becomes relevant in cross-healthcare centres with limited data-sharing policies.

A technological breakthrough, in addition to the proliferation of deep learning in other areas of health, is one of the fundamental tendencies in mammogram analysis. In particular, one of the first tasks in artificial intelligence applications in mammography could be the automatic prediction of healthcare outcomes. Updating the model based on new data input and performing further validation in various clinical settings will likely be key to the model’s success and benefit in clinical settings. Regular updates, continuous monitoring, and model modification in different clinical settings are critical to ensure protocol effectiveness. The results of this study demonstrate that applied deep learning methods to mammogram analysis provide a human-like understanding of mammograms regarding performance, accuracy, and reliability. Therefore, this study lays the groundwork for advancements in developing more deep learning-assisted mammography systems for breast cancer diagnosis and detection across various clinical settings. This can improve the quality of life for breast cancer patients and support working professionals to make the most informed decisions about breast cancer treatment in practice.

Limitations of the study

The study has several limitations. The performance of the P-CycCNN model relies on the dataset’s size and quality, and its ability to extend to new general problems or clinical settings remains unclear. Its training was done only on mammography images; thus, its usage is restricted to this kind of imaging. Due to high complexity and computational requirements, the model may be unsuitable for implementation in low-resource settings, and overfitting can also be a concern. Unclear models of AI could equally negatively impact clinician confidence, while the requirement for frequent revisits and retraining makes scaling much more challenging. Furthermore, the evaluation criteria of the model should be interpreted carefully since they may not depict the clinical outcome, and,other issues such as integration into existing health organisations, data privacy, loss of confidentiality and legal approval may also be a barrier. Ultimately, biases introduced by the dataset may potentially impact the model’s fairness; thus, improvements must be made for even wider applications.

Supplemental Information

Supplemental Information 1 Implementation code.

Supplemental Information 2 README file.

Additional Information and Declarations

Competing Interests

The authors declare that they have no competing interests.

Author Contributions

Sudha Prathyusha Jakkaladiki conceived and designed the experiments, performed the experiments, analyzed the data, performed the computation work, prepared figures and/or tables, authored or reviewed drafts of the article, and approved the final draft.

Filip Malý performed the experiments, analyzed the data, performed the computation work, prepared figures and/or tables, authored or reviewed drafts of the article, and approved the final draft.

Data Availability

The following information was supplied regarding data availability:

The Curated Breast Imaging Subset DDSM Dataset is available at Kaggle:

https://www.kaggle.com/datasets/awsaf49/cbis-ddsm-breast-cancer-image-dataset.

Code is available at Zenodo:

Sudha Prathyusha Jakkaladiki, & Filip Maly. (2025). A Hybrid Deep Learning Approach with Progressive Cyclical CNN and Firebug Swarm Optimization for Breast Cancer Detection. Zenodo. https://doi.org/10.5281/zenodo.15654215.

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
