# Peer review of "A hybrid deep learning approach with progressive cyclical CNN and firebug swarm optimization for breast cancer detection"

_PeerJ Computer Science, doi:10.7717/peerj-cs.3119_

## Round 0.1 · original submission · Major Revisions

**Language Note:** The review process has identified that the English language must be improved. PeerJ can provide language editing services - please contact us at [email protected] for pricing (be sure to provide your manuscript number and title). Alternatively, you should make your own arrangements to improve the language quality and provide details in your response letter. – PeerJ Staff

Reviewer 1 ·

Basic reporting

Novelty and contribution: The manuscript would benefit from clearer positioning in relation to prior work. For example, how does P-CycCNN compare to other adaptive learning rate strategies? Is the use of FSO novel in this context?

Language and grammar: The manuscript contains numerous grammatical errors, awkward constructions, and redundant phrasing. To improve clarity and flow, a professional proofreading service is strongly recommended. Please include a proofreading certificate with your revised submission.

Structure and focus: The manuscript tends to be overly verbose. Several sections (e.g., Introduction, Methods) repeat concepts or overexplain basic elements. Consider streamlining the text to improve readability.

Limitations and future work: The manuscript ends with a reasonable list of limitations. Please consider elaborating on how the model could be adapted for other imaging modalities or implemented in resource-constrained settings.

Experimental design

The pipeline is well described in principle, but several sections are overly detailed while still lacking clarity in key areas. In particular, the progressive learning strategy and cyclical scheduling could be explained more concisely. Some algorithm steps (e.g., overlapping patch segmentation) could be summarized in pseudocode or visual diagrams rather than lengthy paragraphs. Also, the results are reported from a single train/test split (70/30), with no mention of cross-validation, repeated runs, or confidence intervals.

Validity of the findings

The reported performance is strong (98% test accuracy, 0.95 AUC), and the comparison to other CNN variants is appreciated. However, the following points should be addressed:
- Please report standard deviation or variance across multiple runs (e.g., 5-fold cross-validation) to strengthen the evaluation.
- The improvement margin over baseline CNNs is relatively modest. A brief discussion on why the hybrid method improves performance would be helpful (e.g., benefit of combining deep and LBP/shape features).

·

Basic reporting

No Comments

Experimental design

This study proposes a hybrid deep learning approach with progressive cyclical CNN and swarm optimization for breast cancer detection, integrating CNN features extracted from pre-trained models and hand-crafted features to enhance performance. Although the topic is interesting and important in medical field, the paper has major issues, and lacking contributions. Here are my comments:

1. The contributions are missing in abstract with quantitative analysis
2. Major contributions should be listed at the end of introduction
3. Recent articles need to be incorporated in related works
4. et al. need to be changed with italics (eg: Nascimento et al., Sivapriya et al., Amgad et al.)
5. Pre-processing stages not clear. Add block diagram for pre-processing stages and pre-processing results.
6. Tabulate the number of hand-crafted features in mean and standard deviation format
7. Tabulate hyper-parameters used in proposed model
8. Ablation studies of the proposed CNN model need to add
9. Firebug Swarm Optimization (FSO) more explanation needed. How this algorithm improves efficiency of the proposed method
10. In Efficacy Measures:
The test set accuracy was 98. Clarification needed, is this percentage.
11. Reference need to be updated with 2024 and 2025.
12. Compare your results with recent articles.

Validity of the findings

No Comments

Additional comments

No Comments

·

Basic reporting

1. Relevance and Scope
Rating: Very Good
The paper addresses a highly relevant topic in AI-assisted medical diagnostics, specifically breast cancer detection using deep learning. It aligns well with the journal’s scope and contributes to the growing field of medical image analysis.
Suggestion:
Clarify how the proposed method fills a specific research gap not yet addressed in the literature.
Add a clear statement of objectives at the end of the introduction and emphasize how this work differs from or improves upon existing methods.

Experimental design

The methodology is promising but lacks detail. While the architecture and concepts are presented, there is insufficient explanation of:

Experimental setup (e.g., training-validation-test splits)

Hyperparameter settings

Justification for choosing specific models

Implementation details of the hybrid feature fusion

Suggestions:

Include more technical details such as optimizer settings, epochs, loss functions, and computing infrastructure.

Explain why ResNet-50 and VGG16 were selected over other models.

Clarify whether k-fold cross-validation or stratified sampling was used.

Validity of the findings

The concept of combining a cyclical CNN with a novel swarm optimization approach is interesting, but the originality is not strongly emphasized. It is unclear whether Firebug Swarm Optimization is a new or adapted method.

Suggestions:

Highlight the novelty of the FSO method or reference prior applications of FSO if they exist.

Provide more justification for using hybrid feature extraction in this context.

Additional comments

The performance metrics are impressive (98–99% accuracy), but there is a lack of comparative benchmarking with state-of-the-art methods on the same dataset. Moreover, no statistical significance or confidence intervals are provided.

Suggestions:

Include comparisons with recent models or methods in the same domain.

Provide statistical analysis to support performance claims.

Clarify whether the improvement is due to the model architecture, FSO optimization, or both.
Ensure all figures are self-explanatory.

Consider including a system architecture diagram to enhance clarity.

---

## Round 0.2 · accepted · Accept

The reviewers seem satisfied with the recent changes and therefore, I can recommend this article for acceptance.

Reviewer 1 ·

Basic reporting

The authors have addressed all comments. This paper is recommended for publication.

Experimental design

-

Validity of the findings

-